# Countering misinformation via WhatsApp: Preliminary evidence from the COVID-19 pandemic in Zimbabwe

**Jeremy Bowles**[1]ᵒ, **Horacio Larreguy**[1]ᵒ*, **Shelley Liu**[2]ᵒ

**1** Department of Government, Harvard University, Cambridge, MA, United States of America, **2** Goldman School of Public Policy, UC Berkeley, Berkeley, CA, United States of America

ᵒ These authors contributed equally to this work.
* hlarreguy@fas.harvard.edu

**Data Availability Statement:** The full set of replication files is available from Dataverse (doi: 10.7910/DVN/MDF4SO6).

**Funding:** The authors received no specific funding for this work.

## Abstract

We examine how information from trusted social media sources can shape knowledge and behavior when misinformation and mistrust are widespread. In the context of the COVID-19 pandemic in Zimbabwe, we partnered with a trusted civil society organization to randomize the timing of the dissemination of messages aimed at targeting misinformation about the virus to 27,000 newsletter WhatsApp subscribers. We examine how exposure to these messages affects individuals' beliefs about how to deal with the virus and preventative behavior. In a survey of 864 survey respondents, we find a $0.26\sigma$ increase in knowledge about COVID-19 as measured by responses to factual questions. Through a list experiment embedded in the survey, we further find that potentially harmful behavior—not abiding by lockdown guidelines—decreased by 30 percentage points. The results show that social media messaging from trusted sources may have substantively large effects not only on individuals' knowledge but also ultimately on related behavior.

## Introduction

Social media platforms have become a central source of information for individuals in the Global South [1]. For example, since in sub-Saharan Africa traditional media reach is low and mobile data costs to access the internet are high, WhatsApp has become a low-cost "one-stop-shop" [1, 2]. Unfortunately, social media platforms are also highly susceptible to misinformation due to low cost of access, virality of posts, individuals' trust in their social network, and the high cost of fact-checking [3–6]. Amidst the COVID-19 pandemic, as had been the case with the 2014-2015 Ebola epidemic [7] and the 2015-2016 Zika epidemic [8], social media has exacerbated this misinformation problem and muddied public knowledge about the virus throughout the Global South [9–11].

We study whether trusted sources of information can also leverage the ubiquity of social media to combat misinformation and related potentially harmful behavior. Specifically, we examine the effectiveness of WhatsApp messages from a trusted civil society organization (CSO) in Zimbabwe aimed at targeting misinformation in the context of the COVID-19

**Competing interests:** The authors have declared that no competing interests exist.

pandemic. Zimbabweans rely heavily on WhatsApp to access and share information due to prohibitive data costs and the anonymity that WhatsApp affords. As a result, the social network accounts for close to half of all internet traffic in Zimbabwe, far more than competing platforms such as Facebook, which commands only 1% of internet traffic [12]. While the exact number of unique WhatsApp users is not known, an estimate in 2017 suggests that there are at least 5.2 million WhatsApp users in the country [13]. This is roughly 37% of the country's total population or 60% of the country's population over the age of fourteen.

During the study period, the COVID-19 virus had reached Zimbabwe, and the government had just imposed a national lockdown to limit the spread of the virus. Already, across various social media platforms, and particularly through WhatsApp, posts with misinformation about virus transmission and cures had gone viral. Further, due to the low official infection rates, many questioned the necessity of preventative measures [14]. Misinformation about the virus and low trust in the government threatened the likelihood of lockdown compliance in the country.

To combat this problem, we partnered with two organizations, Internews and Kubatana, over a two-week period to disseminate truthful information about COVID-19 in Zimbabwe. Each week, we leverage Kubatana's large and wide-reaching WhatsApp subscriber base to randomize the timing of message dissemination, with the treated condition receiving these messages on Monday while the control group receives messages on Saturday. We measure individuals' knowledge through a mid-week survey, and embed a list experiment designed to measure compliance with social distancing, while addressing concerns relating to demand effects and social desirability bias. Contrary to mixed results from the Global North on the dissemination of health-related misinformation [15–18], we find that social media messaging against misinformation from a trusted source can increase both knowledge about COVID-19 and also preventative behavior. These results speak to the potential of trusted social media sources to combat misinformation and related potentially harmful behavior among its subscribers. However, more work is needed to extrapolate these findings to other sources and samples.

## Materials and methods

We partner with two organizations in Zimbabwe to carry out this study. First, we partnered with Internews, an international non-governmental organization (NGO) operating in Zimbabwe. Internews focuses on training and supporting independent media across the world to help provide people with trustworthy and high-quality information. Our second partner, which implemented the study, is Kubatana, a trusted online media civil society organization (CSO) that was formed in 2001. Kubatana primarily shares information with its subscribers on issues relating to civil and human rights in Zimbabwe through its email, Facebook, Twitter, and WhatsApp channels. The organization began using WhatsApp as a method of distribution in 2013. Today, it has over 27,000 WhatsApp subscribers from across the country divided roughly across 133 WhatsApp broadcast lists. These lists were created based on the month and year of subscription and contain up to 256 subscribers per list.

### Research design

Each week, our two partner organizations jointly crafted a short WhatsApp message (S1 Appendix). In the first week, the message explained COVID-19's rates of transmission and emphasized the importance of social distancing to lower them. In the second week, the message debunked a viral piece of misinformation on fake cures for COVID-19. Kubatana disseminated the messages in English, Shona, and Ndebele, which are the three main languages in

Zimbabwe, through its WhatsApp broadcast lists. In addition, the organization maintained its usual publishing and activity schedule.

To evaluate their effect, we randomized the timing of these messages at the WhatsApp broadcast list level. Subscribers in broadcast lists assigned to the *treatment* condition in a given week were sent the message on Monday, while subscribers in broadcast lists assigned to the *control* condition were sent the message on Saturday. Between these two days of the week, Kubatana sent two additional messages to its subscribers through WhatsApp. First, between Tuesday and Wednesday, it sent its weekly newsletter. Second, on Thursday, it distributed a short survey designed to test treatment effects on 1) knowledge of the information disseminated in the messages, and 2) behavior relating to social distancing. Respondents were given the option of responding to the survey either directly through WhatsApp messages or through the survey platform Qualtrics according to their preference. Notably, Kubatana disseminated both the messages and survey without sharing broadcast list information with us, to avoid potential reputational costs in a context where anonymity is highly valued. Therefore, we had no access to individual identifiers from survey respondents. As we discuss later, this did not affect our results.

This research design has three advantages. First, by randomizing the timing of each message rather than the dissemination itself, all WhatsApp subscribers eventually received important information regardless of their treatment condition. Second, by having Kubatana's weekly newsletter in between the WhatsApp message to treated broadcast lists, we reduced the likelihood that survey respondents would scroll back to a previous message to search for the correct answer in the knowledge-testing questions. Third, by allowing respondents to respond through WhatsApp, we maximized the response rate. In line with our expectation due to the mobile data costs in Zimbabwe, the survey response rate was four times higher through WhatsApp than through Qualtrics.

All research was carried out in compliance with local Zimbabwean research standards and was reviewed to be in accordance with standards set forth by the Committee on the Use of Human Subjects at Harvard University. By randomizing the timing of the messaging rather than whether recipients received messages at all, we did not withhold potentially important information from the sample. Further, because the researchers received no identifiable data on the participants and did not interact with any of Kubatana's subscribers directly, the research was granted exemption status.

## Data

In week 1, Kubatana disseminated the messaging to 13,921 individuals on Monday (treatment condition) and to 13,400 individuals on Saturday (control condition). In week 2, for which treatment assignment was re-randomized, messages were sent to 13,566 individuals on Monday (treatment condition) and 13,755 on Saturday (control condition). This yielded a survey sample comprising of 868 respondents over two weeks, with 585 (2% response rate) from the first week and 283 (1% response rate) from the second week. These response rates are similar to those of other studies where survey respondents are recruited through social media in sub-Saharan Africa [19]. Respondents to the survey are demographically representative: 55% of our survey respondents are male and 76% live in urban localities, aligning with evidence from nationally-representative surveys, which estimate that 59% of frequent social media users in Zimbabwe are male and 69% live in urban areas [20]. Descriptively, a substantial share of respondents report believing in fake cures that have prominently spread through social media. 30% of respondents believe that drinking hot water will cure the virus and 25% believe that inhaling steam will. S1 Table provides descriptive statistics relating to the sample.

We evaluate outcomes relating to *knowledge* and *behavior*. We measured knowledge using a standardized index, or z-score, of responses to factual questions that relate to the message sent in a given week. Directly asking about preventative behavior likely induces demand effects or social desirability bias. Each week, we thus measured behavior using a list experiment, a common technique to estimate the prevalence of sensitive behaviors [21]. Respondents were given a list of activities and asked how many they had performed in the past three days. One version of this list, the *short* experimental list, comprised a list of four non-sensitive activities. The other version of this list, the *long* experimental list, used the same four non-sensitive activities *and added* one sensitive activity—visiting a friend or family member outside of their homes during the mandated nationwide COVID-19 lockdown period. We randomly assigned respondents to a *short* or *long* experimental list at the WhatsApp broadcast list level. A comparison of the reported number of activities, across respondents assigned to 'short' and 'long' experimental lists within *the same* treatment condition (i.e. whether they had received the message on Monday or Saturday of that week), provides an unbiased measure of the prevalence of the sensitive activity among the respondents assigned to a given treatment condition. Comparing this measure *across* treatment conditions then provides an estimate of the effect of the intervention on behavior.

Each week, to assign each WhatsApp broadcast list to a treatment condition, we initially blocked broadcast lists into groups of four, grouping lists which had been created around the same time together. Then, within each block, we randomly assigned one list to each of the four possible combinations of treatment conditions and experimental list length. Such blocking and within-block randomization is commonly done prior to the random assignment of treatment to improve the precision of estimated treatment effects by subsequently including block fixed effects in the estimation [22]. In S2 Table, we show that survey response rates and respondent characteristics are balanced across treatment conditions. This suggests an absence of differential selection into survey participation based on treatment assignment.

We estimate treatment effects on *knowledge* by regressing the z-score index onto a treatment indicator. We estimate treatment effects on *behavior* by regressing the number of activities reported in the list experiment onto a treatment indicator, a long experimental list indicator, and the interaction between the two. We provide specifications with and without controlling for respondent covariates. We include week fixed effects and either randomization block fixed effects or, more demandingly, WhatsApp broadcast list fixed effects. Standard errors are clustered at the level of the WhatsApp broadcast list-week throughout. Further, we explore subgroup treatment effects by splitting our sample across gender, urbanity and week of the intervention. We provide additional information on estimation in S4 Appendix. All statistical analyses were conducted using Stata 16, while graphics are produced in R.

## Results

First, we examine the effects of treatment assignment on respondent knowledge about the information delivered. Fig 1 plots the treatment effects using different permutations of our specifications. The results suggest substantively large effects of the WhatsApp messages on individual knowledge. In the baseline specification with randomization block fixed effects, respondents assigned to a treated WhatsApp broadcast list in a given week report factual knowledge $0.26\sigma$ greater than respondents assigned to a control list ($p < 0.001$). Treatment effects are slightly larger in the specification with WhatsApp broadcast list fixed effects at $0.45\sigma$ ($p < 0.001$). These correspond to roughly 7 percentage points, or 12% increase, in correct responses. Across specifications, results are unchanged by the addition of respondent covariates.

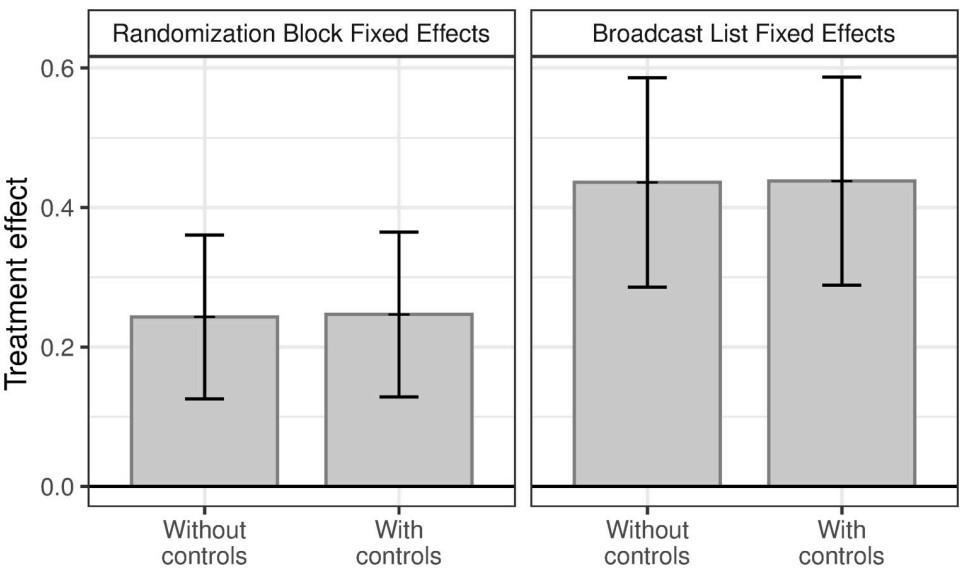

**Fig 1. Treatment effects on knowledge.** Estimates of the treatment effect of WhatsApp messages on a standardized index of responses to factual questions that relate to the messages sent. 95% confidence intervals plotted. All specifications include week fixed effects. Standard errors clustered at the week-broadcast list level.

Second, we examine treatment effects on respondents' preventative behavior. Fig 2 plots the treatment effects using different permutations of our specifications. In the baseline specification, *among respondents assigned to the control condition*, 37% ($p < 0.001$) did not comply with social distancing. However, *among respondents assigned to the treatment condition*, this behavior drops to 7% ($p = 0.47$). The difference between these effects is statistically significantly different ($p < 0.05$), implying that the WhatsApp messages changed related behavior. Estimated

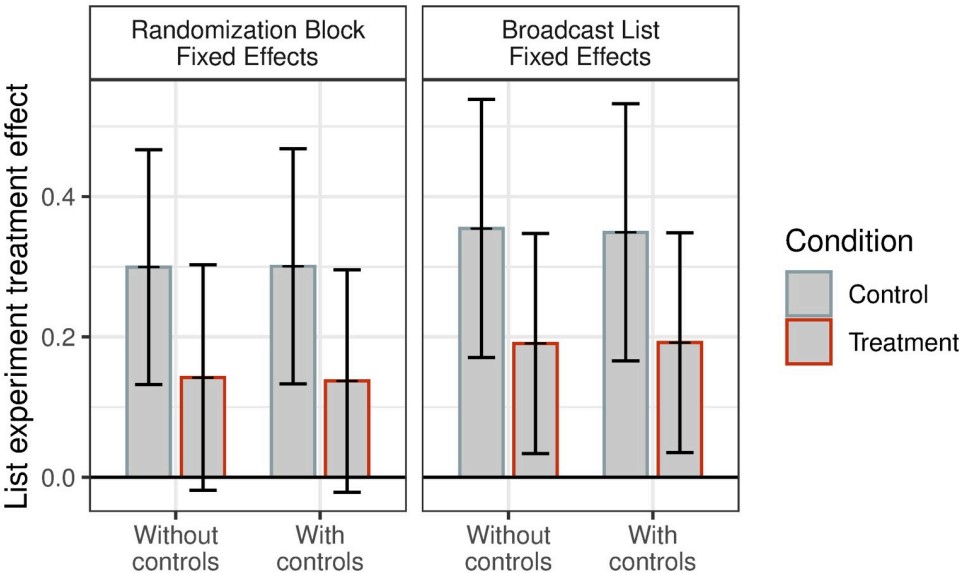

**Fig 2. Treatment effects on behavior.** Estimates of the treatment effect of WhatsApp messages on behavior measured through a list experiment between subscribers in treated and control broadcast lists. 95% confidence intervals plotted. All specifications include week fixed effects. Standard errors clustered at the week-broadcast list level.

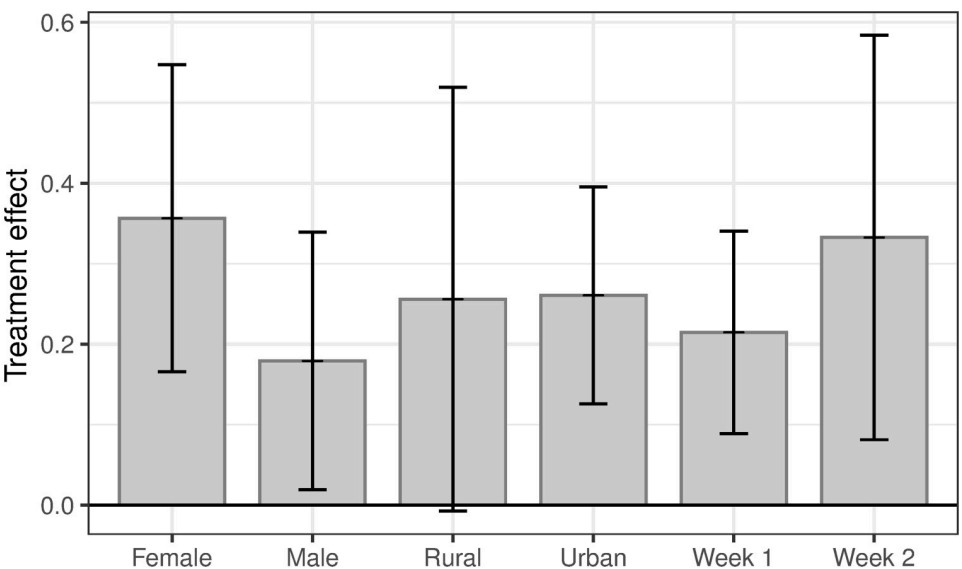

**Fig 3. Subgroup treatment effects on knowledge.** Estimates of the treatment effect of WhatsApp messages on a standardized index of responses to factual questions that relate to the messages sent. 95% confidence intervals plotted. All specifications include randomization block fixed effects and (apart from by-week estimates) week fixed effects. Standard errors clustered at the week-broadcast list level.

treatment effects are again slightly larger when using WhatsApp broadcast list fixed effects and are robust to the addition of respondent covariates. The magnitudes of these treatment effects are comparable to those from other studies seeking to facilitate healthy behavior in the Global South [23]. Importantly, due to the use of a list experiment, these treatment effects on behavior cannot be explained by demand effects and social desirability bias, or respondents scrolling back to a previous message to search for the correct answer. The consistency of the effects on behavior with the effects on knowledge, which are potentially affected by such possible biases, helps to bolster confidence in the results overall.

Lastly, we examine subgroup treatment effects on the two outcomes in Figs 3 and 4 based on gender, rurality, and week of intervention. We find relatively uniformly estimated effects across subgroups. While statistically insignificant, treatment effects on knowledge among women are greater than among men ($p = 0.25$), while effects on behavior are not different between women and men ($p = 0.85$). We also find similar treatment effects in Weeks 1 and 2. S3, S4 and S5 Tables provide the estimated regression coefficients to support the figures.

## Discussion

In sum, our results indicate encouraging positive changes in knowledge and behavior among WhatsApp subscribers of a trusted source. While WhatsApp has been identified as a platform through which misinformation easily spreads, we show that trusted CSOs can also leverage WhatsApp's reach to successfully get individuals to reassess their misconceptions and correct related behavior. This effect is roughly similar across the urban-rural as well as the gender divide, highlighting the power of WhatsApp messages from a trusted source to counter misinformation. These findings, then, stress the potential of CSOs in sub-Saharan Africa to fight misinformation. They further highlight the similar role that other WhatsApp newspapers in the region might play (e.g., The Continent in South Africa and 263Chat in Zimbabwe).

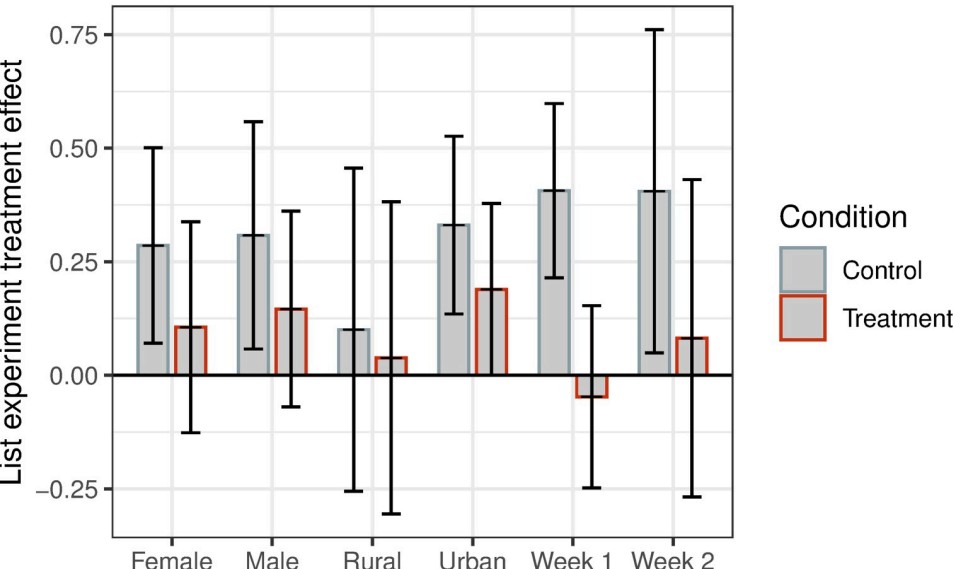

**Fig 4. Subgroup treatment effects on behavior.** Estimates of the treatment effect of WhatsApp messages on behavior measured through a list experiment between subscribers in treated and control broadcast lists. 95% confidence intervals plotted. All specifications include randomization block fixed effects and (apart from by-week estimates) week fixed effects. Standard errors clustered at the week-broadcast list level.

The study's context and findings contribute to recent work on the effectiveness of messages to correct misinformation across a variety of issues ranging from health to politics [15, 18, 24]. These studies present mixed findings and are particularly negative with respect to vaccination campaigns [16, 17]. However, most them provide evidence from lab and online experiments in the Global North, while far fewer studies take place in the Global South. Similarly, there is a dearth of field experimental evidence in this context, which is likely to be most informative for scaling up related policies [25, 26]. Our positive findings from a field experiment in Zimbabwe suggest that there are especially high returns to correcting misinformation, especially surrounding ongoing health crises where people are uncertain and seeking information [7, 27, 28].

Our results may deviate from those in prior scholarship in part due to the population that we study: individuals who have already self-selected into receiving information from a human rights NGO. While the sample appears demographically similar to the broader population of social media users in Zimbabwe, the subscribers are likely already receptive to information delivered to them by Kubatana and hence one should be cautious when extrapolating the treatment effects we find to other samples. However, our sample represents an important, and growing, population in the developing world—of individuals who are exposed to misinformation through social media, but also seek independent, credible sources of information through well-established NGOs.

As part of our ongoing surveying efforts in Zimbabwe, we asked respondents for the sources of COVID-19 information that they trust the most. Descriptively, we find that citizens are most likely to trust an international organization first, followed closely by local NGOs or CSOs, and third by a message that mentions a news source (see Fig 5).

In conjunction with the experimental results we present above, this evidence suggests that a trusted source of information can use the same social media channels to disseminate information that both combats misinformation and changes related behavior. Thus, while we caution generalizing our results to general public in Zimbabwe, our results speak specifically to the important

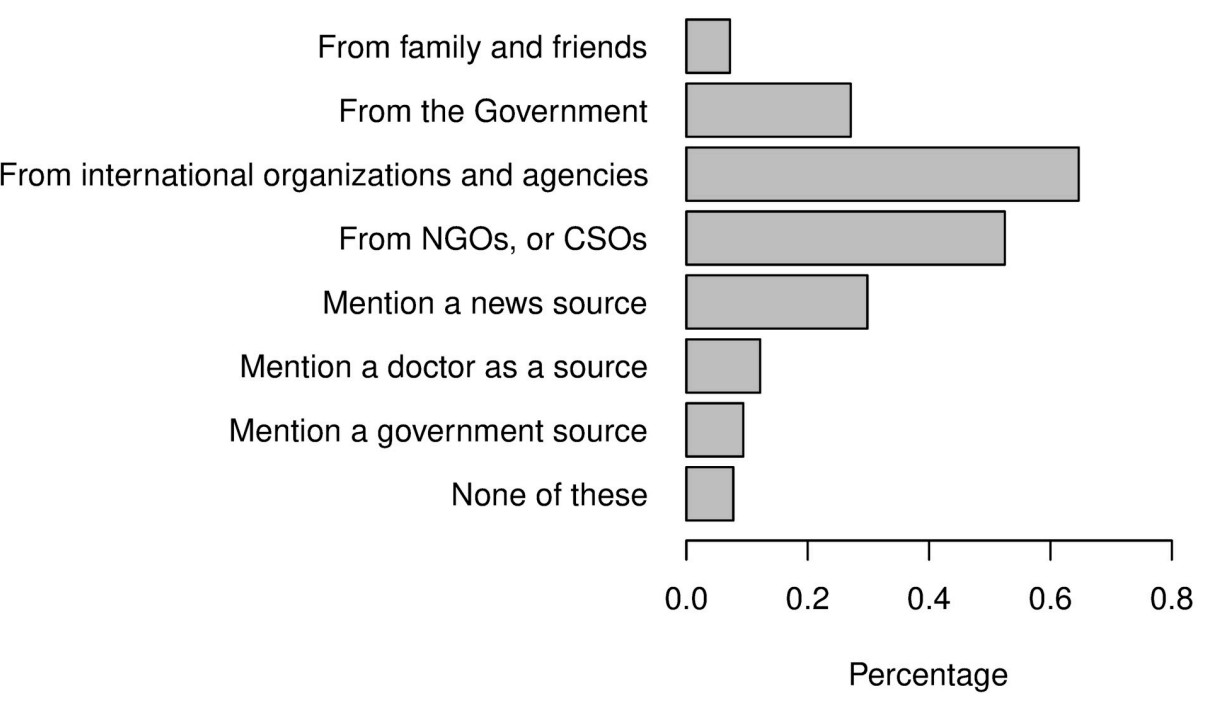

**Fig 5. Trusted sources of information about COVID-19.** Respondents were asked to select up to three sources of information that they trust most on WhatsApp to deliver information about COVID-19.

role that trusted sources play, particularly in confusing informational situations such as health crises [29], and in an authoritarian context where trust in information might be low [30]. Existing scholarship emphasizes the importance of how information is framed [31], and the credibility of the information source for the recipient [32]. During the COVID-19 pandemic, the identification and dissemination of correct information represent an important challenge. While fact-checking can contribute to a source's credibility [33], particularly during emergency situations, it might be outpaced by the spread of misinformation through social media [34, 35].

Future research should consider how best to integrate social media messaging aimed at targeting misinformation into CSOs' ongoing programming, while at the same time highlighting their relative importance. During the study, Kubatana's WhatsApp messaging increased three-fold, from one WhatsApp message a week. Even after two weeks, the organization reported four unsubscribers—a number that, while low, is highly unusual for it. Moreover, in the second week, there was a 50% drop in survey responses relative to the first week. Additional work on identifying how to maximize the benefits of such messaging without inducing disengagement will be of great importance for devising a sustainable way to counter misinformation in the Global South.

## Supporting information

**S1 Appendix. Messages.**
(PDF)

**S2 Appendix. Coding decisions.**
(PDF)

**S3 Appendix. Survey questions used.**
(PDF)

**S4 Appendix. Estimation.**
(PDF)

**S1 Table. Summary statistics.**
(PDF)

**S2 Table. Balance.**
(PDF)

**S3 Table. Knowledge.**
(PDF)

**S4 Table. Behavior.**
(PDF)

**S5 Table. Outcomes by week.**
(PDF)

## Acknowledgments

With thanks to our partnering organizations, Internews and Kubatana, for their cooperation. Fotini Christia and Kevin Croke provided useful comments. IRB exemption granted by Harvard Committee on the Use of Human Subjects (IRB20-0602).

## Author Contributions

**Conceptualization:** Jeremy Bowles, Horacio Larreguy, Shelley Liu.

**Data curation:** Jeremy Bowles, Horacio Larreguy, Shelley Liu.

**Formal analysis:** Jeremy Bowles, Horacio Larreguy, Shelley Liu.

**Investigation:** Jeremy Bowles, Horacio Larreguy, Shelley Liu.

**Methodology:** Jeremy Bowles, Horacio Larreguy, Shelley Liu.

**Project administration:** Jeremy Bowles, Horacio Larreguy, Shelley Liu.

**Supervision:** Horacio Larreguy.

**Visualization:** Jeremy Bowles, Horacio Larreguy, Shelley Liu.

**Writing – original draft:** Jeremy Bowles, Horacio Larreguy, Shelley Liu.

**Writing – review & editing:** Jeremy Bowles, Horacio Larreguy, Shelley Liu.

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
