## [Decision Letter · Decision Letter 0]

1 Sep 2020

PONE-D-20-23677

Countering misinformation via WhatsApp: Evidence from the COVID-19 pandemic in Zimbabwe

PLOS ONE

Dear Dr. Larreguy,

Thank you for submitting your manuscript to PLOS ONE. After careful consideration, we feel that it has merit but does not fully meet PLOS ONE’s publication criteria as it currently stands. Therefore, we invite you to submit a revised version of the manuscript that addresses the points raised during the review process.<please by="" manuscript="" revised="" submit="" your="">

Please include the following items when submitting your revised manuscript:</please>

We look forward to receiving your revised manuscript.

Kind regards,

Khin Thet Wai, MBBS, MPH, MA (Population & Family Planning Resear

Academic Editor

PLOS ONE

Journal Requirements:

2. Your ethics statement must appear in the Methods section of your manuscript. If your ethics statement is written in any section besides the Methods, please move it to the Methods section and delete it from any other section. Please also ensure that your ethics statement is included in your manuscript, as the ethics section of your online submission will not be published alongside your manuscript.

3. Please upload a copy of Supporting Information Tables S1-S5 which you refer to in your text.

4. We note you have included a table to which you do not refer in the text of your manuscript. Please ensure that you refer to Tables 1-5 in your text; if accepted, production will need this reference to link the reader to the Table.

Additional Editor Comments (if provided):

This research provides the sound evidence critical to improve countermeasures for the infodemic of COVID-19 and to impede with correct information through social media platforms. It covered the demographically representative sample.

Specifically, authors could improve the integrity of this research by expanding/adding/clarifying the following issues apart from responding to reviewers.

(1) To expand the abstract by adding the concrete results with relevant statistics;

(2) LINE 83-84: Survey response rates were compared between Whatsapp and Qualtrics. As such, the authors needed to add the total number of subscribers to Qualtrics.

(3) LINE 103-104: To clarify 'short experimental list'; 'long experimental list'.

(4) LINE 116-118: To clarify "initially blocked broader list into groups of four" and cite a reference for this method.

(5) LINE 121-124: To add the software used for regressing the treatment effects.

Reviewers' comments:

Reviewer's Responses to Questions

**Comments to the Author**

1. Is the manuscript technically sound, and do the data support the conclusions?

Reviewer #1: Yes

Reviewer #2: Yes

2. Has the statistical analysis been performed appropriately and rigorously? 

Reviewer #1: Yes

Reviewer #2: I Don't Know

3. Have the authors made all data underlying the findings in their manuscript fully available?

Reviewer #1: Yes

Reviewer #2: Yes

4. Is the manuscript presented in an intelligible fashion and written in standard English?

Reviewer #1: Yes

Reviewer #2: Yes

5. Review Comments to the Author

Reviewer #1: Methodology thoroughly explained. It is necessary to include what proportion of internet/social media users are regularly use Whatapps, in comparison to other social media app, to judge the generalization of findings of study. If the author describes It is good if more explicit data for comparison between arms of study can be described.

Reviewer #2: The paper experiments on how to combat misinformation by disseminating truthful information about COVID-19 in Zimbabwe. The author examined how exposure to these messages can affect individual belief in preventing behavior. To assess that effect, the authors experimented with the timing when the messages were sent.

The methodology sounds solid, but I missed some numbers in the paper: the number of users in each group (treatment and control) in the Whatsapp and the percentage of people in the lists who respond to the surveys. It is not clear with there is an overlap of respondents between the two surveys (first and second weeks)? If so, how much the first survey influenced the second study in the second week?

I am not sure if the number of individuals is enough to conclude the findings. But,

I am not a statistician and only can trust the reported results.

Finally, the authors did not mention demand bias. How many respondents change their behavior or opinions as a result of being part of a study? The authors identified themselves in the surveys as being part of Harvard University. What would happen with the results of the survey if the survey was not identified or identified as being from Kubatana organization?

6. PLOS authors have the option to publish the peer review history of their article (what does this mean?). If published, this will include your full peer review and any attached files.

Reviewer #1: No

Reviewer #2: No

---

## [Editor Report · Decision Letter 1]

18 Sep 2020

Countering misinformation via WhatsApp: Preliminary Evidence from the COVID-19 pandemic in Zimbabwe

PONE-D-20-23677R1

Dear Dr. Larreguy,

We’re pleased to inform you that your manuscript has been judged scientifically suitable for publication and will be formally accepted for publication once it meets all outstanding technical requirements.

Kind regards,

Khin Thet Wai, MBBS, MPH, MA (Population & Family Planning Resear

Academic Editor

PLOS ONE

Additional Editor Comments (optional):

All comments are addressed satisfactorily.
---

## [Editor Report · Acceptance letter]

23 Sep 2020

PONE-D-20-23677R1 

Countering misinformation via WhatsApp: Preliminary Evidence from the COVID-19 pandemic in Zimbabwe 

Dear Dr. Larreguy:

I'm pleased to inform you that your manuscript has been deemed suitable for publication in PLOS ONE. Congratulations! Your manuscript is now with our production department. 

Kind regards, 

on behalf of

Dr. Khin Thet Wai 

Academic Editor

PLOS ONE